# Exploring the Role of Osteosarcoma-Derived Extracellular Vesicles in Pre-Metastatic Niche Formation and Metastasis in the 143-B Xenograft Mouse Osteosarcoma Model

**DOI:** 10.3390/cancers12113457

**Published:** 2020-11-20

**Authors:** Alekhya Mazumdar, Joaquin Urdinez, Aleksandar Boro, Matthias J. E. Arlt, Fabian E. Egli, Barbara Niederöst, Patrick K. Jaeger, Greta Moschini, Roman Muff, Bruno Fuchs, Jess G. Snedeker, Ana Gvozdenovic

**Affiliations:** 1Department of Orthopedics, Balgrist University Hospital, CH-8008 Zurich, Switzerland; Alekhya.Mazumdar@balgrist.ch (A.M.); jurdinez.unq@gmail.com (J.U.); alexandarboro@yahoo.com (A.B.); MArlt@gmx.net (M.J.E.A.); Roman.Muff@balgrist.ch (R.M.); fuchs@sarcoma.surgery (B.F.); Jess.Snedeker@balgrist.ch (J.G.S.); 2Laboratory for Orthopedic Biomechanics, Institute for Biomechanics, ETH Zurich, CH-8008 Zurich, Switzerland; fabian.egli@hest.ethz.ch (F.E.E.); barbara.niederoest@hest.ethz.ch (B.N.); patrick.jaeger@hest.ethz.ch (P.K.J.); greta.moschini@hest.ethz.ch (G.M.)

**Keywords:** osteosarcoma, premetastatic niche, extracellular vesicles, myeloid cell infiltration

## Abstract

**Simple Summary:**

Osteosarcoma is an aggressive bone cancer that frequently metastasizes to the lungs and is the second leading cause of cancer-associated death in children and adolescents. Therefore, deciphering the biological mechanisms that mediate osteosarcoma metastasis is urgently needed in order to develop effective treatment. The aim of our study was to shed light on the primary tumor-induced changes in the lungs prior to osteosarcoma cell arrival using a xenograft osteosarcoma mouse model. Furthermore, we investigated the functional role of osteosarcoma-derived extracellular vesicles in pre-metastatic niche formation and metastasis. We showed that the primary tumor initiates an influx of CD11b^+^ myeloid cells in the pre-metastatic lungs. Furthermore, we demonstrated that osteosarcoma-derived extracellular vesicles alone can recapitulate myeloid cell infiltration in the lungs of naïve mice, but are insufficient to promote osteosarcoma metastasis. Our findings provide valuable insight into the field of osteosarcoma-derived extracellular vesicles and their role in pre-metastatic niche formation in the 143-B osteosarcoma model.

**Abstract:**

The pre-metastatic niche (PMN) is a tumor-driven microenvironment in distant organs that can foster and support the survival and growth of disseminated tumor cells. This facilitates the establishment of secondary lesions that eventually form overt metastasis, the main cause of cancer-related death. In recent years, tumor-derived extracellular-vesicles (EVs) have emerged as potentially key drivers of the PMN. The role of the PMN in osteosarcoma metastasis is poorly understood and the potential contribution of osteosarcoma cell-derived EVs to PMN formation has not been investigated so far. Here, we characterize pulmonary PMN development using the spontaneously metastasizing 143-B xenograft osteosarcoma mouse model. We demonstrate the accumulation of CD11b^+^ myeloid cells in the pre-metastatic lungs of tumor-bearing mice. We also establish that highly metastatic 143-B and poorly metastatic SAOS-2 osteosarcoma cell-derived EV education in naïve mice can recapitulate the recruitment of myeloid cells to the lungs. Surprisingly, despite EV-induced myeloid cell infiltration in the pre-metastatic lungs, 143-B and SAOS-2 EVs do not contribute towards the 143-B metastatic burden in the context of both spontaneous as well as experimental metastasis in severe-combined immunodeficient (SCID) mice. Taken together, OS-derived EVs alone may not be able to form a functional PMN, and may perhaps require a combination of tumor-secreted factors along with EVs to do so. Additionally, our study gives a valuable insight into the PMN complexity by providing the transcriptomic signature of the premetastatic lungs in an osteosarcoma xenograft model for the first time. In conclusion, identification of regulators of cellular and molecular changes in the pre-metastatic lungs might lead to the development of a combination therapies in the future that interrupt PMN formation and combat osteosarcoma metastasis.

## 1. Introduction

Osteosarcoma is the most common primary malignant bone tumor in children and adolescents with a high propensity for pulmonary metastases [1]. As effective treatment options for metastatic disease are currently lacking, metastases are the primary cause of death in osteosarcoma patients. Hence, a deeper understanding of the metastatic process is required to successfully develop anti-metastatic therapeutic strategies, critical for combating patient mortality.

Metastasis is a complex multistep process and it is becoming clear that successful metastasis is not solely dictated by the autonomous properties of tumor cells such as genetic mutations or epigenetic regulation, but also by their complex interplay with host components as well as the landscape of the metastatic microenvironment [2,3,4,5]. It is now well-established that primary tumors release factors that orchestrate the preparation of the local parenchyma at future metastatic sites prior to the seeding of cancer cells and thereby regulate the formation of a specialized microenvironment, designated as the pre-metastatic niche (PMN) [6]. The establishment of pulmonary PMNs includes a series of molecular and cellular changes that lead to increased vascular permeability, reprogramming of stromal cells, extracellular matrix (ECM) remodeling, recruitment of several cell populations and lung inflammation [7,8,9]. These tumor-driven stromal alterations provide a hospitable microenvironment that supports the survival and outgrowth of disseminated tumor cells leading to metastatic colonization. In addition to tumor-secreted soluble factors, tumor-derived extracellular vesicles (EVs), such as exosomes, have been identified as crucial determinants of PMN and metastasis in multiple cancer types [10]. Exosomes are 30–150 nm vesicles of endosomal origin, that carry bioactive cargo such as mRNA, miRNA and proteins. Transfer of tumor-derived exosomal cargo has been instrumental to mediate tumor cells—stromal cells communication leading to PMN formation. For example, in melanoma, EV-mediated transfer of the receptor tyrosine kinase MET was shown to be essential for homing of bone-marrow progenitor cells to the lungs and initiation of the PMN [11]. Transfer of miR-105 to endothelial cells in the lungs caused destabilization of tight junctions and increased vascular permeability and lung metastasis of breast cancer cells [12]. Similarly, transfer of miR-122 to lung fibroblasts was shown to reduce their glucose uptake, leading to increased nutrient availability for disseminated breast cancer cells [13].

Research exploring the functional role of EVs for osteosarcoma tumor progression in vitro has accelerated over the last decade [14]. Proteomic analysis of multiple osteosarcoma cell-line-derived EVs revealed that they contain proteins associated with tumor progression, angiogenesis, cell adhesion, and migration [15]. Moreover, evidence gathered from several in vitro studies implicates that osteosarcoma-derived EVs are involved in the intercellular communication between tumor cells and the neighboring tumor/non-tumor cells in the local bone tumor microenvironment. For example, osteosarcoma EVs contain pro-osteoclastogenic cargo or specific miRNAs that induce murine osteoclast formation and stimulate their bone resorption activity [16,17]. EVs derived from metastatic osteosarcoma cells have also been shown to promote the in vitro migration and invasion of human osteoblast cells [18]. In a previous study, we demonstrated that osteosarcoma cell-derived EVs can induce lung fibroblast reprogramming in vitro and direct fibroblast activation and differentiation towards a myofibroblast/cancer-associated fibroblast phenotype through EV-associated TGFβ1 and SMAD2 pathway activation [19]. In contrast to significant progress being made to understand the function of osteosarcoma cell-derived EVs using in vitro approaches, only a few studies employed an in vivo strategy [14,20]. Although osteosarcoma cell-derived EVs have been shown to preferentially localize to the lungs of mice after intravenous injections [21], their potential involvement in PMN formation as well as their direct effects on metastatic properties of osteosarcoma cells in an in vivo setting have not been addressed so far. Furthermore, a better understanding of the PMN and defining stromal alterations in the lung tissue prior to the arrival of tumor cells might open new avenues for therapeutic interventions.

In this study, we first define the metastasis kinetics of 143-B cells in the well-established xenograft orthotopic osteosarcoma model. Upon determination of the pre-metastatic phase, we identify the primary tumor-induced changes in the pre-metastatic lungs by using next-generation RNA-sequencing (RNA-seq) of lungs harvested from tumor-bearing and naïve mice. Ultimately, we explore the role of osteosarcoma cell-derived EVs in mediating PMN formation and metastasis in both spontaneous and experimental mouse models of osteosarcoma.

## 2. Results

### 2.1. Pre-Metastatic Phase Determination in the Orthotopic 143-B Osteosarcoma Model in SCID Mice

In order to elucidate the effects of primary tumor cells on the lung tissue prior to metastatic cell arrival, we first set out to determine the duration of the pre-metastatic phase in our established xenograft spontaneous metastasis model of osteosarcoma. We therefore performed a time-course study that examined pulmonary micro-and macro-metastases formation over time upon orthotopic transplantation of 143-B cells into SCID mice. The 143-B cells were transduced with *LacZ*, *luciferase*, and *mCherry* (hereafter 143-B cells), which enabled us to accurately visualize and quantify the progression of metastatic disease [22]. Mice were sacrificed at indicated time intervals after intratibial 143-B cell inoculation (Figure 1A). As expected, primary tumor development monitoring through bioluminescent imaging and caliper measurements revealed exponential tumor growth (Figure 1B,C). Intriguingly and to our surprise, upon ex vivo X-gal staining of whole-mount lungs, high numbers of indigo-blue tumor cells could be detected 4 h after tumor cells were injected into the tibia implicating that tumor cells enter the bloodstream upon intratibial injection (Appendix A). However, and more importantly, the lungs were cleared of tumor cells 4 days after tumor cell injection (TCI) (Appendix A) indicating that the leaked cells were unable to survive in the lungs due to the absence of a metastasis-supportive environment or PMN, underlining the importance of the PMN formation for the subsequent development of metastases. Subsequently, the very first micro-metastases were detectable at day 12 and the first macro-metastases were detectable only at day 17. The number of metastases gradually increased over time until the end of the experiment (Figure 1D,E).

Hence, we could establish that the pre-metastatic phase in the 143-B model lasts approximately 12 days after orthotopic TCI, which is then succeeded by the metastatic colonization phase. These observations imply that the primary tumor drives PMN formation in the distant lungs enabling successful colonization of tumor cells 12–17 days after intratibial TCI.

### 2.2. 143-B Osteosarcoma Primary Tumors Drive Immune Cell Infiltration into the Pre-Metastatic Lungs

To investigate the molecular changes that occur in the PMN, we performed mRNA profiling of the PMN by next generation RNA sequencing of pre-metastatic bulk lung tissue isolated from tumor-cell injected animals at day 12 and lung tissue of PBS-injected control animals. The divergence of gene signatures between the two groups during an unsupervised hierarchical clustering analysis provides evidence of PMN establishment in our model (Appendix A). The filtering of differentially expressed genes revealed 810 upregulated genes and only 210 downregulated genes, indicating that the PMN fostered an activated milieu (Figure 2A). Gene ontology (GO) analysis of the top 250 upregulated genes demonstrated a significant association with immune response and reprogramming of metabolic processes, similar to findings in other tumor models (Appendix A) [9,13]. Moreover, we also observe processes in response to hypoxia, alteration of lipid metabolism and oxidative stress, which have all been linked with myeloid cell mobilization and immunosuppressive functions [23,24]. Hence, considering “inflammatory response” (GO:0006954) in the PMN as a central process, we present the upregulated expression of 19 genes annotated under it (Figure 2B). Involved genes included chemoattractants such as *S100a8*, *S100a9*, *CxcL13*, *Il1b*, and *Ccl24*, as well as immune receptors such as *Il1rl2*, *Ccr1*, *Cd14*, and *Cd163*. Since CD11b^+^ cells have previously been identified to be the most common sub-population of myeloid cells infiltrating the PMN, we performed immunofluorescence staining to explore if the upregulation of chemoattractants in the lungs is associated with CD11b^+^ cell recruitment [25]. Our results confirmed the accumulation of CD11b^+^ myeloid cells in the lungs of tumor-bearing mice at both a pre-metastatic time point (day 12) and late metastatic time point (day 28) (Figure 2C). Moreover, we do not observe any changes in CD11b^+^ myeloid cell population in the lungs of PBS injected control mice through the course of the experiment when compared to naïve animals (Appendix A).

In summary, through RNA-seq we show that the 143-B primary tumor induces a pro-inflammatory milieu in the pre-metastatic lungs. GO analysis reveals characteristic features previously reported in the PMN of other tumor types, such as reprogramming of metabolic states and oxidative stress. Through immunofluorescence we demonstrate the recruitment of inflammatory CD11b^+^ cells to the pre-metastatic as well as metastatic lungs in tumor-bearing animals.

### 2.3. Osteosarcoma Cell-Derived EV Education Drives Inflammatory Myeloid Cell Infiltration into Lungs of Naïve Mice

We next postulated that primary tumor-derived EVs may contribute to the PMN formation and tumor-driven inflammatory myeloid cell infiltration in pre-metastatic lungs. We tested our hypothesis in an experimental model of “in vivo EV education” which involved successive intravenous injections of tumor cell line-originating EVs into naive SCID mice (Figure 3A). Our aim was to compare effects of EVs isolated from 143-B osteosarcoma cells with high metastatic potential with effects of EVs isolated from SaOS-2 cells that have a low metastatic potential. EVs used in the in vivo study were shown to have typical exosomal features in our previous study [19]. Three groups of animals were treated with the vehicle control (PBS) or with 10 µg of either 143-B or SAOS-2 EVs twice a week for three weeks and their lungs were analyzed for CD11b^+^ cell infiltration. Results pooled from two independently performed experiments showed increased *Itgam* (integrin alpha M gene encoding CD11b) mRNA expression in the lungs of 143-B and SAOS-2 EV-treated mice compared to the control (Figure 3B). Additionally, through both flow cytometry (Figure 3C) and immunofluorescent staining (Figure 3D), we demonstrate an increase in CD11b^+^ cells in the lungs of EV-treated mice in independent experiments. Co-staining with granulocytic marker, Gr-1 showed that the majority of CD11b^+^ cells were double-positive for Gr-1 (Appendix A), indicating that these cells are most likely neutrophils or alternatively granulocytic myeloid-derived suppressor cells [26]. CD11b^+^ cells and Gr-1^+^ cells were normalized to the total number of DAPI^+^ cells for comparison (Appendix A).

These results demonstrate that both 143-B and SAOS-2-cell-derived EV education mobilizes CD11b^+^ Gr-1^+^ immune cells to the lungs of naïve mice, recapitulating the effects of a primary tumor in mice.

### 2.4. Osteosarcoma Cell-Derived EV Education Does Not Alter the Primary Tumor Growth and Metastasis of Osteosarcoma Cells in a Spontaneous Metastatic 143-B Mouse Model

To further examine the effects of osteosarcoma cell-derived EV education on primary tumor growth and metastasis, we intratibially injected mice with 143-B cells after three weeks long treatment with either vehicle, 143-B or SAOS-2 EVs (Figure 4A). Additionally, mice were treated with vehicle or EVs for 1 week after TCI. Pre-treatment with EVs did not influence primary tumor growth kinetics, the final primary tumor volume or the extent of bone osteolysis (Figure 4B). Furthermore, although we demonstrated the presence of CD11b^+^ Gr-1^+^ myeloid-derived cells in the pre-metastatic lungs, which is associated with increased tumor metastasis in other cancer models [27], we did not observe any significant differences in the number of X-gal stained micro-and macro-metastases among the different treatment groups (Figure 4C). Taken together, we show that both 143-B and SAOS-2 cell-derived EV education do not affect 143-B primary tumor growth or metastasis in an orthotopic experimental setup.

### 2.5. Osteosarcoma Cell-Derived EV Education Does Not Alter the Experimental Metastatic Potential of the 143-B Cells

Using an orthotopic setup closely resembles the metastatic disease in human patients and allows for the investigation of different steps in the metastatic cascade. However, the presence of a primary tumor may lead to the secretion of alternative factors and EVs, potentially masking the effects of EV pre-treatment. Taking this into account, to investigate the potential direct effects of EVs on tumor cell lung colonization as the final step of metastasis, we utilized an experimental metastasis model and intravenously injected 143-B cells into mice pre-treated for three weeks with either vehicle, 143-B or SAOS-2 EVs, following which animals were further treated with vehicle or EVs for a week after TCI (Figure 5A). In vivo bioluminescent imaging of luciferase activity was performed to measure metastatic burden, which demonstrated no significant differences between the different treatment groups (Figure 5B). Nineteen days after TCI, the total numbers of X-gal stained micro-and macro-metastases remained unaffected by EV education (Figure 5C).

Additionally, in a complementary approach investigating EV effects on the total metastatic load, we determined the total metastatic area covered by all metastatic nodules through histological examination of lung sections in a subset of mice (Figure 5D). In line with our in vivo observations, H&E staining revealed similar metastatic areas in the control and 143-B experimental groups. Interestingly, we observed a decrease in metastatic area in SAOS-2 EV treated mice. This observation is supported by a previous study that reports decreased metastases when pre-treated with EVs isolated from non-metastatic melanoma cells, however, we only report a decrease in metastatic area and not the number of metastases [28]. In summary, we establish that both 143-B and SAOS-2 cell-derived EV education do not affect metastases formation in an experimental metastasis model in SCID mice.

## 3. Discussion

In recent years, substantial effort has been directed at understanding the interaction between osteosarcoma tumors and the distant metastatic microenvironment. However, the critical events driving the metastatic colonization remain poorly defined. In this study, we explore PMN formation in the 143-B osteosarcoma tumor model and demonstrate that the pre-metastatic phase lasts for about twelve days after orthotopic implantation of 143-B cells. Additionally, to our knowledge, ours is the first study to report unbiased RNA-seq data that characterizes the PMN in osteosarcoma. Our RNA-seq data demonstrates characteristic features of the PMN in the pre-metastatic lungs of tumor-bearing mice described in other tumor models, including reprogramming of the metabolic processes and influx and mobilization of CD11b^+^ myeloid cells [13,29]. We additionally confirm that inflammatory cell infiltration in the lungs can be phenocopied through osteosarcoma-cell-derived EV education in naïve mice, in line with the findings reported in pancreatic, melanoma, and breast cancer-derived EV education [9,30,31]. 143-B EVs have previously been demonstrated to promote metastasis in an indirect manner, by intratibially co-injecting in vitro EV-treated mesenchymal stem cells along with 143-B tumor cells [20]. In contrast to the mentioned study, we utilize a direct in vivo EV education approach and in vivo and ex vivo imaging modalities to visualize spontaneous and experimental metastasis. We, however, failed to observe any significant differences in the metastatic burden upon treatment with both poorly metastatic SAOS-2 and highly metastatic 143-B osteosarcoma-EVs when compared to the control in both an orthotopic as well as an experimental metastasis model in SCID mice. EV education alone in this particular osteosarcoma model is not sufficient to promote tumor cell dissemination and lung colonization and points to the additional prerequisite tumor-secreted factors.

Despite being heavily reported to play a critical role in preparing the PMN and promoting the seeding of metastatic cells [29,32], it is now becoming clear that bone marrow-derived CD11b^+^ tumor-associated myeloid cells may exist in different stages of activation/differentiation, and consequently are either pro-tumorigenic [33] or anti-tumorigenic [34]. For example, persistent activation of signal transducer and activator of transcription 3 (STAT3) signaling in myeloid cells was shown to be crucial for their pro-metastatic function, and interruption of this signaling axis led to the abrogation of PMN development [35]. Myeloid cells also form a heterogeneous population, of which only a subset of cells, such as myeloid-derived suppressor cells (MDSCs) are responsible for promoting tumor development [36]. Chemical/peptibody-mediated perturbation of such pro-metastatic MDSCs was shown to disrupt PMN formation, reduce metastatic burden, and prolong overall survival in various mouse models [33].

In osteosarcoma, the literature provides evidence that myeloid cells in the lungs play an important role in either promoting or inhibiting osteosarcoma progression. Namely, CD11b^+^/CXCR4^+^ myeloid cell accumulation in the primary tumor was correlated with reduced cytotoxic T-cell activity and poor overall survival in patients [37]. Additionally, tumor-infiltrating myeloid cells were documented to be critical drivers of radiation-induced osteosarcomagenesis in mice [38]. Osteosarcoma-secreted angiopoietin like 2 (ANGPTL2) was recently demonstrated to recruit neutrophils in pre-metastatic lungs, and Ly6G mediated depletion of these neutrophils resulted in decreased spontaneous metastasis in syngeneic osteosarcoma models [39]. Anti-tumorigenic effects have also been reported where the type 2 diabetes drug, metformin was shown to induce anti-tumor effects through the metabolic reprogramming of CD11b^+^ myeloid cells, causing them to inhibit osteosarcoma tumor growth [40]. However, in our present study, it is currently unclear if the majority of EVs-recruited myeloid cells observed in the lungs are pro-tumorigenic or anti-tumorigenic CD11b^+^ cells or if there is a mix of both populations. State-of-the-art techniques such as mass-cytometry and/ or single cell RNA-seq of sorted CD11b^+^ cells might be elegant approaches to gather valuable information about the phenotypic differences between the various subpopulations of recruited and/or resident cells. Non-metastatic melanoma-derived EV education has been previously reported to induce a pro-phagocytic differentiation of macrophages, leading to the inhibition of lung colonization, while metastatic melanoma-derived EVs promoted lung colonization in both syngeneic and xenograft mouse melanoma models [28]. Hence, it was unexpected to see that both 143-B EV-and SAOS-2 EV-treatment showed no difference in the total number of metastasis, when compared to the PBS-treated control. Our results implicate that, although osteosarcoma cell-derived EVs lead to stromal changes in the pre-metastatic lung, these alterations are not sufficient to result in an increased metastatic burden. We speculate that a combination of EVs and other tumor-secreted factors may be required to drive the formation of a complete and functional PMN. Secretion of soluble factors such as VEGF-A, TNFα, and TGFβ from melanoma and lung cancer models has been previously demonstrated to induce the expression of inflammatory chemoattractants such as S100A8 and S100A9 in the premetastatic lungs, leading to myeloid cell recruitment [29]. Interestingly, we also observe an increase of S100A8 and S100A9 expression in the pre-metastatic lungs of osteosarcoma-bearing mice. Additionally, high VEGF and TGFβ expression have been associated with a poor prognosis among patients with osteosarcoma [41,42], and interruption of VEGF and TGFβ signaling was shown to disrupt osteosarcoma progression and metastasis [42,43]. Other soluble factors, such as IL-8 and ANGPTL2 have been additionally shown to mediate osteosarcoma metastatic progression and PMN formation in different mouse models [39,44]. These reports collectively provide evidence that tumor-secreted factors are also involved in osteosarcoma PMN formation, but further investigation is still required to help understand if they act in combination with EVs to establish a functional PMN. Challenging mice with CRISPR-Cas9 knockout tumors for specific secreted factors in combination for knockout for Rab27a/b might help us further dissect the underlying mechanism of PMN formation.

Although the 143-B mouse model closely recapitulates disease progression in patients and is a very commonly used model to study osteosarcoma metastasis [20,45], the findings of the study presented here, need to be confirmed in additional xenograft models and using alternative EV treatment protocols. Moreover, the functional relevance of osteosarcoma-derived EVs needs to be investigated in an immune-competent context using syngeneic mouse models. Evading immune-surveillance through macrophage/MDSC-mediated suppression of anti-tumor CD8^+^ and CD4^+^ T-cells is a common strategy used by tumors to promote metastasis [46,47]. Additionally, osteosarcoma-derived EVs have already been shown to promote M2 polarization of alveolar macrophages and blunt their tumoricidal function, indicating their immune-suppressive properties [48]. Therefore, further efforts are required to understand the role of EVs in the immune escape of tumors.

Osteosarcoma cell-derived EVs have been previously identified to localize to the lungs after intravenous injections [21]. Moreover, the extent of EV accumulation in other organs is currently uncertain. With the advent of novel state of the art in vivo EV visualization systems, future studies should be performed to look at osteosarcoma EV uptake in various non-pulmonary microenvironments and assess their impact on PMN formation and metastasis development in these alternative microenvironments. This may provide valuable evidence on the mechanism of action of these EVs in vivo.

Another important aspect to consider is the high tumor heterogeneity seen in osteosarcoma. This leads to numerous ways of endowing metastatic competence and, therefore, molecular profiling of patient-derived EVs will be valuable in the future [49,50]. In addition to this, proteomic profiling of EVs obtained from liquid biopsies from osteosarcoma patients will lead to the discovery of valuable biomarkers for early detection of cancer in patients. Advances have been made in this direction with, Hoshino et al. recently publishing the proteomic profile of EVs isolated from multiple patient-derived tumor explants and plasma [51]. This includes a small cohort of osteosarcoma patient-derived EVs, where the authors identified 6 osteosarcoma EV-associated proteins that successfully distinguished osteosarcoma patients from healthy controls. They included alpha skeletal muscle actin (ACTA1), gamma-enteric smooth muscle actin (ACTG2), ADAMTS13, hepatocyte growth factor activator (HGFAC), neprilysin, and tenascin C. Further effort is required to validate these biomarkers and discover novel biomarkers in the liquid biopsies from a larger cohort of osteosarcoma patients in order to improve plasma-based screening and diagnosis in patients.

In conclusion, by describing a transcriptomic signature of the pre-metastatic lung, our study provides an insight into the complexity of the PMN together with a platform for discovering essential PMN-associated factors that regulate lung colonization. Future studies investigating the function of the described changes together with the cross-linking of the relevant EV-associated molecular cargo will enable a better understanding of the metastatic process in osteosarcoma.

Finally, identifying the crucial regulatory factors in the osteosarcoma-derived secretome responsible for the essential cellular and molecular changes in the pre-metastatic lungs might lead to development of combination therapies that interrupt PMN formation and combat osteosarcoma metastasis.

## 4. Materials and Methods

### 4.1. Cell Culture

The human osteosarcoma cell lines SaOS-2 (HTB-85) and 143-B (CRL-8303) cells were obtained from ATCC (Gaithersburg, MD, USA). In order to enable visualization of tumor cells within mouse tissues, in vivo and ex vivo, 143-B cells were sequentially transduced with the *LacZ* gene, *mCherry* gene and *Firefly luciferase* gene as described recently (hereafter referred to as 143-B cells) [52,53,54]. Osteosarcoma cells were cultured in 1:1 DMEM (4.5 g/L glucose) and HamF12 medium (61965026 and 1765029, Thermo Fisher Scientific, Paisley, UK) and supplemented with 10% heat-inactivated fetal bovine serum (FBS, 10500, Thermo Fisher Scientific, Paisley, UK), hereafter referred to as complete medium. The cells were grown in 5% CO_2_ and 95% air in humidified conditions at 37 °C. Cells were confirmed to be mycoplasma negative and authenticated using short tandem repeat DNA profiling (Microsynth, Balgach, Switzerland) and compared to the German Collection of Microorganisms and Cell Cultures database.

### 4.2. EV Isolation, Fibercell Hollow-Fiber Bioreactor Culture and Bioreactor EV Purification

Extracellular vesicles (EV) were isolated from either 2D dish culture (henceforth referred to as 2D EVs) or 3D bioreactor culture (henceforth referred to as 3D EVs) as described previously [19]. For 2D cultures, OS cells were seeded into 10 × 15 cm dishes in complete medium. Upon reaching 70–80% confluency, the cells were washed with PBS and subsequently incubated in serum-free medium (SFM) for 48 h. EVs were isolated from the conditioned medium using standard sequential ultracentrifugation and sterilized by passing through a 0.2 μm Steriflip^®^ pore filters (SCGP00525, Millipore-Sigma, Burlington, MA, USA). Alternatively, for 3D cultures, a medium-sized, hollow-fiber culture cartridge, with a 20 kDa molecular weight cut-off (C2011, Fibercell Systems, Frederick, MD, USA) was set up according to the manufacturer’s instructions. Harvests from the bioreactor were collected 3–5 times per week, depending on the glucose concentration of the medium reservoir. EVs were isolated using standard sequential ultracentrifugation and purified on a 30% sucrose-tris-D2O cushion [55]. Purified EVs were resuspended in PBS and sterilized by passing through 0.2 μm Steriflip^®^ pore filters. The protein content in EV preparations was determined with a BCA Protein Assay Kit (23225, Thermo Fisher Scientific, Paisley, UK). The EVs were adjusted to a concentration of 10 µg/100 µL and stored at −80 °C until use in experimental procedures. We previously reported the characterization of EVs through nanoparticle tracking analysis, negative staining transmission electron microscopy and flow cytometry analysis of EV-coated beads [19]. The data pertaining to the characterization of EVs were used in these studies are presented in Appendix A.

### 4.3. Animal Care and Orthotopic Induction of 143-B Osteosarcoma Tumors in SCID Mice

All experiments including animals were approved by the Ethics Committee of the Veterinary Office of the Canton Zurich and was conducted in accordance with the Swiss Animal Protection Law. Eight to ten-week-old female severe combined immunodeficiency mice (SCID) (CB17/Icr-Prkdc^scid^/IcrIcoCrl, Charles River Laboratories, Sulzfeld, Germany) were maintained in enriched, individually ventilated cages with 12 h/12 h light/dark cycles. Mice were acclimatized for 10–14 days without any interventions after transportation. For the pre-metastatic niche characterization study, mice were randomized based on weight into 2 groups for tumor cell injections (TCI) or control PBS injections. Mice were anesthetized as described before [54], and the medullary cavity of the left proximal tibia was injected with either 10 μL of 10^5^ 143-B cells in PBS/0.05% EDTA or 10 μL of PBS/0.05% EDTA on day 0. Primary tumor development was measured at the time of sacrifice by both bioluminescence imaging as well as caliper measurements of the length and the width of the tumor leg (t) and the non-injected control leg (c). Leg volume (V) was calculated with the formula V = 0.5 × length × (width)^2^ and primary tumor volume = Vt–Vc. Routine checks accessing pain, discomfort, limping, and weight loss were used to determine the start of pain treatment in mice. Once mice started limping due to the tumor burden, 0.1 mg/kg of intraperitoneal Buprenorphine (Temgesic; Reckitt Benckiser, UK) was administered 2–3 times daily. Tumor-injected groups (6 mice per group) and PBS injected groups (4 mice per group) were then subsequently sacrificed at day 0 (4 h after TCI), day 4, day 7, day 12, day 17, day 21 and day 28 after TCI. Mice were sacrificed according to the euthanasia protocol as described by previously [56].

### 4.4. Ex Vivo Visualization of LacZ-Transduced Metastatic Tumor Cells in the Lung

After sacrifice, the primary tumors and in situ perfused lungs from mice were harvested as reported before [56]. Lungs were fixed for 30 min at RT in 2% formaldehyde and X-gal stained (ALX-582-002-G001, Enzo Life Sciences, Farmingdale, NY, USA). Indigo-blue stained pulmonary metastases on the surface of whole-mount lungs were counted at 4× magnification under the Nikon Eclipse E600 microscope (Nikon Instruments Inc., Amstelveen, Netherlands). Micro-metastases were defined as indigo-blue stained foci smaller than 0.1 mm in diameter, and larger foci were defined as macro-metastasis.

### 4.5. RNA Sequencing, Data Analysis and qRT-PCR

RNA isolation and analysis were performed, as described previously [57]. Briefly, lung tissue from tumor-bearing mice at day 12 (*n* = 6; T1–T6) and PBS injected control mice at day 12 (*n* = 4; PBS1-PBS4) were either snap-frozen directly in liquid nitrogen or embedded in OCT (81-0771-00, Biosystems AG, Muttenz, Switzerland) and then snap-frozen on dry ice. RNA was isolated from either frozen lung tissue (homogenized with the TissueLyser LT (85600, Qiagen, Hilden, Germany)) or cryogenic sections using the RNeasy Mini Kit (74104, Qiagen, Hilden, Germany) and on-column DNase-treatment (79254, Qiagen, Hilden, Germany) according to the manufacturer’s instructions. Total RNA expression analysis and library preparation was performed with a TruSeq Stranded Total RNA kit (20020596, Illumina, San Diego, CA, USA). Paired-end sequencing of 100-nucleotide sequences was performed on an Illumina HiSeq4000 as a service by the Functional Genomics Center Zurich (FGCZ) (http://www.fgcz.ch/). Bioinformatic analysis was performed with the SUSHI analysis framework using edgeR [58,59]. Reads were aligned using the STAR aligner46 of at least 30 bp matching and acceptance at most five mismatches, and at most 5% of mismatches were used. Read alignments were only reported for reads with less than 50 valid alignments. Python 3.7.7 was used together with the pandas (1.0.5), seaborn (0.10.1), and fastcluster (1.1.26) packages for unsupervised clustering of gene signatures [60,61,62]. The correlation metric used to investigate differentially regulated genes contained a significance threshold of *p* < 0.01, fold change >1.5 and false discovery rate (FDR) <0.2. Metascape was used for gene ontology enrichment analysis on the top 250 upregulated genes [63].

For qRT-PCR, samples were processed as reported earlier [19]. Relative expression levels were calculated by using the 2^(−ΔCT)^ method, normalized to *Gapdh*. The primers used were mouse *Gapdh* (forward: 5′-GCC TTC CGT GTT CCT AC-3′; reverse 5′-CCA AGA TGC CCT TCA GTG-3′) and mouse *Itgam* primers (forward: 5′-GGT CTT TGG ATT GAT GCA GAA-3′; reverse: 5′-CCA ACC AGT GTA TAA TTG AGG C-3′).

### 4.6. EV Education and Orthotopic/Experimental Metastasis Induction of Osteosarcoma Tumors in SCID Mice

To analyze the role of EV education in PMN formation, EVs from multiple harvests were pooled together and adjusted to a concentration of 100 µg/mL to avoid batch-specific variation in EVs. Mice were injected with 10 μg of 143-B EVs or SaOS-2 EVs two times a week for 3 weeks. The concentration and treatment schedule were chosen based on the existing literature [13,64]. An equivalent volume of PBS was injected into the control mice. This is defined as the EV education phase for all the involved experiments. Mice were then euthanized, and their lungs were perfused and used for flow cytometry analysis or immunofluorescent staining. Both 2D and 3D EVs were used in two independent experiments, and no difference in outcome was observed between the two sources of EVs. Alternatively, after the EV education phase, mice were further injected with 10 μg of 143-B EVs or SaOS-2 EVs for 1 week after TCI to ensure that EV mediated changes in the PMN were not lost.

To study the role of EV education in primary tumor growth and spontaneous metastasis, 8-to 10-week-old SCID/CB17mice were then intratibially injected with 10 μL of 10^5^ 143-B cells in PBS/0.05% EDTA after 3D EV education. Primary tumor development and osteolysis were examined weekly by Caliper measurements or by X-ray radiographs with a Faxitron^®^ MX-20 DC Digital Radiography System (Faxitron X-Ray Corporation, Tuscon, AZ, USA). Five weeks after TCI, mice were euthanized and lungs were perfused, harvested, fixed and X-gal stained Micro-and macro-metastases were counted.

To analyze the role of EVs in lung colonization experiments, 8-to 10-week-old SCID/CB17 mice were then intravenously injected with 5 × 10^5^ 143-B cells in 100µL of HBSS through the tail-vein after 2D EV education. Weekly measurements of tumor cell burden were performed by measuring luciferase bioluminescence activity. Due to time constraints, 8 mice were randomly chosen per treatment group and followed through the course of the experiment. Briefly, mice were intraperitoneally injected with 100 mg/kg of body weight of XenoLight D-Luciferin (SL30101-1g, SwissLumix, Lausanne, Switzerland) 5 min before imaging. They were then anesthetized with isoflurane (Forane^®^, AbbVie Inc., North Chicago, IL, USA) (5% isoflurane for induction and 2% for maintenance) and bioluminescence was measured using an IVIS Lumina XR imaging system (Caliper Life Sciences Inc., Waltham, MA, USA) and quantified with Living Image Version 3.1 software (Xenogen Corporation, Alameda, CA, USA). The animals were euthanized 19 days after TCI. The lungs were perfused, fixed for X-gal staining, and metastases were quantified.

### 4.7. Flow Cytometry Analysis of Lung Tissues

Shortly after euthanasia, lungs of the 143-B tumor-bearing mice were perfused with PBS, minced and digested in 2 mg/mL of Collagenase D and A (11088866001 Roche, Basel, Switzerland and 10103586001, Sigma-Aldrich, St. Louis, MO, USA) for 1 h at 37 °C. A single-cell suspension was prepared by passing the semi-digested tissue through a 26G needle 2–3 times and then subsequently passing it through a 40 μm cell strainer. Erythrocytes were lysed using ACK-lysing buffer (A1049201, Thermo Fisher Scientific, Paisley, UK) according to the manufacturer’s instructions. Cells were resuspended in staining buffer with TruStain FcX™ PLUS Fc block (156603, Biolegend, London, UK), and incubated on ice for 10 min. The cells were incubated with anti-CD11b (14-0112-82, eBioscience, San Diego, CA, USA; dilution 1:100) for 30 min on ice. Cells were subsequently stained with Alexa Fluor 546TM tagged secondary antibody (A-11081, Thermo Fisher Scientific, Paisley, UK; dilution 1:300) for 30 min on ice and were analyzed using a FACSAriaII (BD, San Jose, CA, USA) and analyzed by FlowJo v10.5.3 software (BD Biosciences, Ashland, OR, USA).

### 4.8. Immunoflurescent and Histological Tissue Analysis

Lung tissue was embedded in OCT mounting medium (81-0771-00, Biosystems Switzerland AG, Muttenz, Switzerland) and frozen using dry ice. Eight µm sections of the tissue were made using an MX35 blade (3051835, Thermo Fisher Scientific, Paisley, UK) on a Leica CM3050-S cryostat-microtome (Biosystems Switzerland AG, Muttenz, Switzerland) at −20 °C. Samples were stained with the following primary antibodies, anti-CD11b (14-0112-81, eBioscience, San Diego, CA, USA; dilution 1:200) and anti-GR-1 (108419, BioLegend, London, UK; dilution 1:200). Alexa Fluor^®^ 568 coupled antibodies (A11081, A10042, Invitrogen, Carlsbad, CA, USA; dilution 1:500) were used as the secondary. Samples were counter-stained using NucBlue™ (R37605, Invitrogen Carlsbad, CA, USA; dilution 1:1000). Images were obtained using a Nikon A1 HD25/A1R HD25 confocal microscope (Nikon Healthcare Japan Inc., Tokyo, Japan) using the NIS-Elements software (Version 5.21.02, Nikon Healthcare Japan Inc., Tokyo, Japan). Images were obtained by independently stitching 4 images acquired at 10x magnification to capture maximum available tissue area. Images were analyzed using ImageJ Version 1.53c (https://imagej.net/).

For histological sections, X-gal stained lung tissue was paraffin-embedded as described previously [65]. Two to three 8 µm sections were collected at 40 μm intervals (6 times) from each lung on SuperFrost Plus™ slides (10149870, Thermo Fisher Scientific, Paisley, UK) and dried on a hot plate at 37 °C. Sections were dewaxed twice using Roticlear^®^ (Carl Roth GmbH & Co., Karlsruhe, Germany), twice with 100% ethanol, twice with 96% ethanol, and then with 70% ethanol before finally being rehydrated using deionized H_2_O. Hematoxylin (9281.1000, Artechemis AG, Zofingen, Switzerland) and eosin (71304, Thermo Fisher Scientific, Paisley, UK) staining was performed according to the manufacturer’s instructions. Images were taken using an EVOS™ XL core microscope (AMEX1100, Thermo Fisher Scientific, Paisley, UK) and analyzed using ImageJ v1.53c software.

### 4.9. Statistical Analysis

Statistical analyses were performed using GraphPad Prism 8 (Version 8.4.2, GraphPad Software, San Diego, CA, USA). The statistical significance of differences between groups was determined using the student’s *t*-test (unpaired, two-tailed) or a one-way ANOVA test with a Bonferroni post-hoc test. Results are shown as mean ± standard error of the mean (SEM). The results were considered significant when *p* < 0.05.

## 5. Conclusions

In conclusion, this study characterizes PMN formation in the orthotopic 143-B osteosarcoma tumor model for the first time and demonstrates a significant increase in transcriptional activity in the PMN through bulk RNA sequencing. The work presented here reveals that osteosarcoma-derived EVs drive myeloid cell recruitment in the PMN. However, through two carefully designed in vivo studies, we establish that both 143-B and SAOS-2-derived EV education alone is not sufficient to promote tumor cell dissemination and lung colonization in a xenograft osteosarcoma model and implicates that additional tumor-secreted factors are required. Identification of EV-associated as well as EV-independent factors contributing to PMN together with the understanding of molecular alterations in the PMN might ultimately open new avenues in osteosarcoma management.

## Figures and Tables

**Figure 1 cancers-12-03457-f001:**
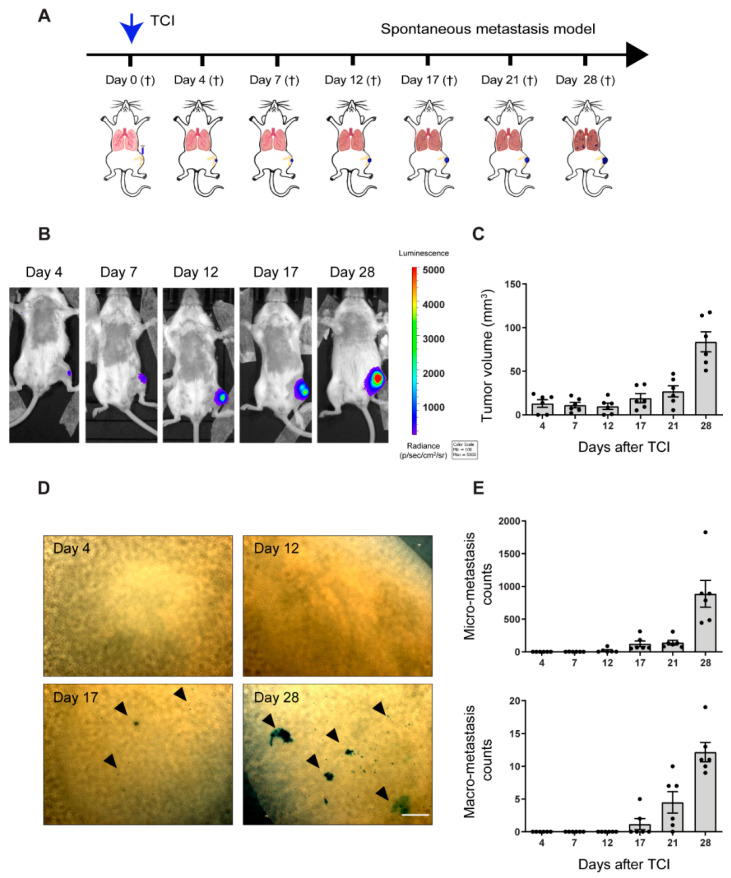
Characterization of spontaneous metastasis in an orthotopic 143-B osteosarcoma model. (**A**) Scheme of the experimental setup, with abbreviations: tumor cell injection (TCI); euthanasia (†). (**B**) Representative IVIS bioluminescence images of tumor-bearing animals sacrificed at the indicated time points. (**C**) Primary tumor growth of 143-B cells over time, monitored by caliper measurements of the tumor volume at indicated time points. (**D**) Representative images of X-gal stained metastases on lung surface of mice sacrificed at indicated time points after TCI. Scale bar, 500 µm. (**E**) Quantification of pulmonary micro-metastasis (<0.1 mm in diameter, top panel,) and macro-metastasis (>0.1 mm in diameter, bottom panel) in mice sacrificed at indicated time points. *n* = 6 per time point.

**Figure 2 cancers-12-03457-f002:**
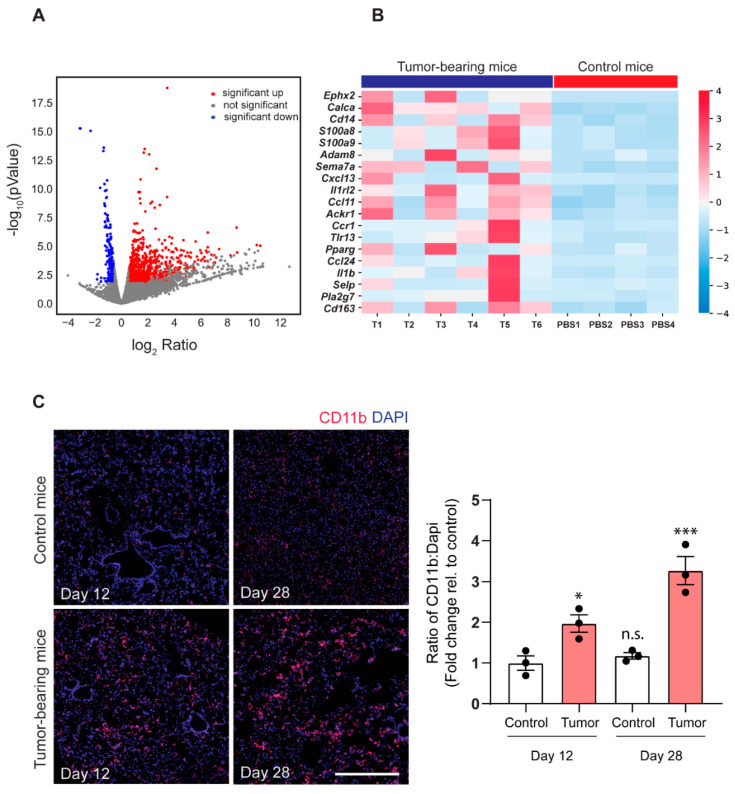
Transcriptomic profiling of pre-metastatic lungs demonstrates an inflammatory response and myeloid cell infiltration. (**A**) Volcano plot of differentially expressed genes in pre-metastatic lungs compared to control lungs. Dots colored in red or blue represent upregulated and downregulated genes, respectively, meeting the thresholds of fold change >1.5, *p*-value < 0.01, and FDR <0.2 respectively. (**B**) Heatmap representation of upregulated genes under the GO term “inflammatory response” (GO:0006954) in tumor-bearing mice vs. control mice. Each column represents one biological replicate (*n* = 6 in the tumor group (T1–T6), *n* = 4 in the control group (PBS1–PBS4)). Cut-off values of fold change are >1.5 and FDR <0.2. The clustering separates the genes by color with positive or negative row-scaled Z-scores represented in red and blue, respectively. (**C**) Representative immunofluorescence images (left panel) and quantification (right panel) of CD11b^+^ cells (red) and nuclei (blue) in lungs of control mice (day 12 and day 28) and tumor-bearing mice (day 12 and day 28), respectively. Scale bar, 500 µm. (n.s., non-significant, *, *p* < 0.05, ***, *p* = 0.001; Bonferroni’s one-way ANOVA test).

**Figure 3 cancers-12-03457-f003:**
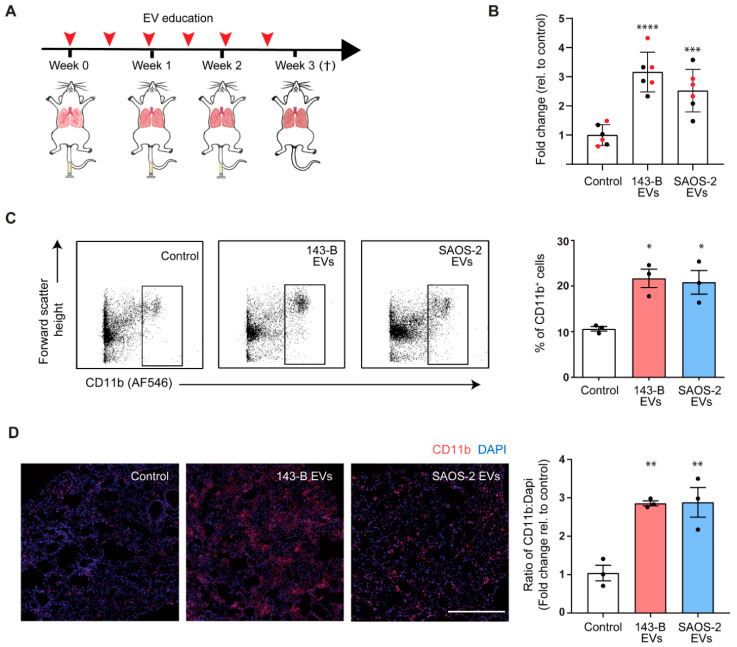
Osteosarcoma cell-derived extracellular-vesicles (EVs) promote accumulation of bone marrow-derived cells in the lungs. (**A**) Schematic illustration of the experimental setup. Mice were treated with either PBS, 143-B EVs or SAOS-2 EVs (10 µg) 2 times a week for 3 weeks after which mice were euthanized. (**B**) Relative mRNA expression levels of integrin alpha M gene encoding CD11b (*Itgam*) in bulk lung RNA in mice of indicated treatment groups. Transcript levels are normalized to *Gapdh* and expressed as fold change relative to control mice. *n* = 6, mice were pooled from two independent experiments indicated in red and black (**C**) Representative flow cytometric profile (left panel) and quantification (right panel) of CD11b^+^ cells isolated from lungs of mice from indicated treatment groups; *n* = 3 mice per group. (**D**) Representative immunofluorescence images (left panel) and quantification (right panel) of CD11b^+^ cells (red) and nuclei (blue) in lungs of mice from indicated treatment groups. Scale bar, 500 µm. (*, *p* < 0.05, **, *p* < 0.01, ***, *p* = 0.0001, ****, *p* < 0.0001, Bonferroni’s one-way ANOVA test).

**Figure 4 cancers-12-03457-f004:**
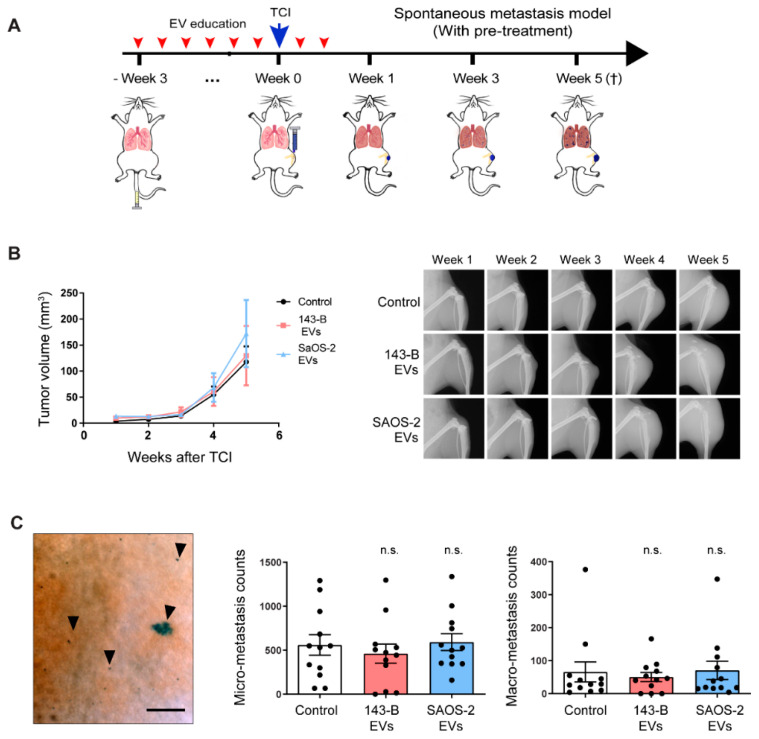
Pre-treatment with osteosarcoma cell-derived EVs does not affect primary tumor growth and spontaneous metastasis in an orthotopic 143-B model. (**A**) Scheme of the experimental setup. (**B**) Primary tumor growth monitored by caliper measurements of the tumor volume at indicated time points (left panel) and representative X-ray images of tumor bearing legs (right panel). (**C**) Representative image of pulmonary metastasis (left panel). Scale bar, 500 µm. Quantification of X-gal stained micro-metastases and macro-metastasis (middle and right panels) on lung mounts of treated mice sacrificed 5 weeks after tumor cell injection; *n* = 12 mice per group. (n.s., non-significant; Bonferroni’s one-way ANOVA test).

**Figure 5 cancers-12-03457-f005:**
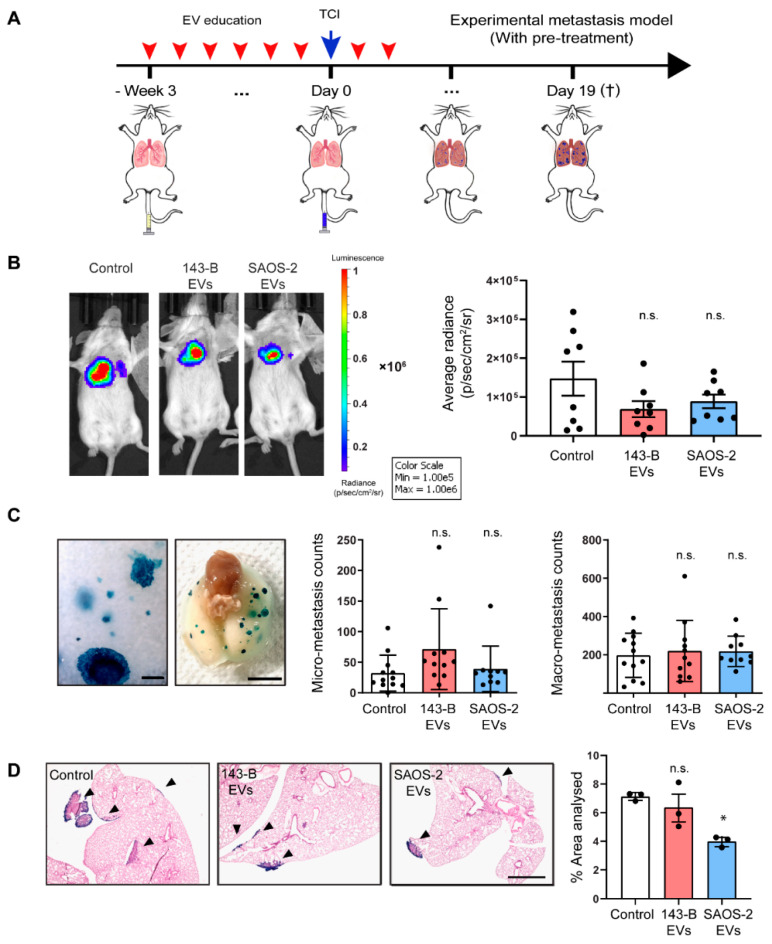
Pre-treatment with osteosarcoma-derived EVs does not affect 143-B experimental metastasis. (**A**) Scheme of the experimental setup. (**B**) Representative IVIS bioluminescence images of tumor cells 15 days after intravenous TCI (top panel) and respective quantification of bioluminescence (bottom panel). *n* = 8 mice for all groups. (**C**) Representative micrographic and macrographic images of 143-B experimental lung metastasis. Scale bars, 250 µm and 0.5 cm respectively. Quantification of X-gal stained micro-metastases (middle panel) and macro-metastases (right panel) on lung mounts of treated mice sacrificed 19 days after tumor cell injection; *n* = 12 mice for the control group, *n* = 11 mice for the 143-B group, *n* = 10 for the SaOS-2 group. (**D**) Representative images of OCT-embedded lung tissue stained for X-gal and H&E from the indicated groups (left panel). Scale bar, 1 mm. Quantification of tumor tissue area (right panel); *n* = 3 mice for all groups. (n.s., nonsignificant; *, *p* < 0.05, Bonferroni’s one-way ANOVA test).

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
