# Peer review of "Exploring the Role of Osteosarcoma-Derived Extracellular Vesicles in Pre-Metastatic Niche Formation and Metastasis in the 143-B Xenograft Mouse Osteosarcoma Model"

_cancers, 2020, doi:10.3390/cancers12113457_

Round 1

Reviewer 1 Report

In the manuscript entitled "Exploring the role of osteosarcoma-derived extracellular vesicles in pre-metastatic niche formation and metastasis in the 143-B xenograft mouse osteosarcoma model" The authors found osteosarcoma derived extracellular vesicles promote influx of CD11b+ myeloid cells in lung but can not promote lung metastasis. Although this phenotype is interesting, it might be difficult to conclude the roles of extracellular vesicles in osteosarcoma metastasis in this study. There are several concerns that need to be addressed to insist what they have claimed in their manuscript.

Major concerns;

  1. To investigate the role of extracellular vesicles in osteosarcoma metastasis, I recommend the authors to establish the 143-B cell line, whose secretion of extracellular vesicles is inhibited, and evaluate its metastatic property in xenograft models. If the establish line could metastasize to lung and promote influx of CD11b+, the results of this study might not closely recapitulate disease progression.

  1. The authors found the interesting phenotypical changes, but I wonder how this identification of phenotypical change contribute to patients’ treatment?

  1. In figure 5D, the authors examined only 3 cases in each group. I recommend additional experiment to evaluate the difference.

Reviewer 2 Report

The study by Mazumdar A, et al was focused on studying the role of extracellular vesicles (EV) in formation of pre-metastatic niche and metastasis in Osteosarcoma. They utilized the 143-B (highly metastatic) xenograft mouse model to demonstrate the accumulation of CD11b+ myeloid cells in pre-metastatic lungs. This is the first study to present the transcriptomic profile of pre-metastatic lungs in a xenograft model of osteosarcoma. Additionally, this study also showed that EVs-derived from osteosarcoma cell lines are inadequate to promote pre-metastatic niche or metastasis formation in severe –combined immunodeficient (SCID) mice.

The strength of this paper is the application of an in vivo strategy using ‘in vivo EV education’ to study the potential involvement of EVs in pre-metastatic niche formation and metastasis development. However, there are some major and minor concerns with the study that are enlisted below:

Major comments:

  1. The authors have demonstrated in Figure 1 that the pre-metastatic phase in the 143-B model lasts approximately 12 days, which is succeeded by metastatic colonization. However, for the ‘in vivo education’ experiments treatment included approximately 3 weeks and then myeloid cell infiltration was quantified. Is there a particular reason why the authors decided to evaluate a 3 week EV treatment regime?
  2. The study makes use of EVs derived from 2D and 3D cultures. But they failed to explicitly explain the source of EVs used in different experiments. In lines 442 and 443, the authors claim that no difference in outcome was observed between the two sources of EVs. It is suggested that the authors should provide data to corroborate this claim and possibly include it as a supplementary figure in the paper.
  3. The authors conclude that the findings provide a better understanding of osteosarcoma progression, however the major conclusion from this study is rather that EVs do not promote osteosarcoma premetastatic niche formation in this model. The major conclusion should be revised to reflect this.
  4. Additionally, the authors should include data pertaining to the characterization of EVs that were used in these studies as a supplemental figure instead of citing a previous publication.
  5. In Figure 2, data for control mice on Day 28 is missing. It will be helpful to include quantification for the immunofluorescence images as well.
  6. For statistical analysis of data with multiple groups, the authors have utilized one-way ANOVA with Bonferroni’s posthoc test which is unnecessarily conservative (with weak statistical power; PMID: 30157585). Is there a particular reason that the authors did not employ Tukey’s test for posthoc comparisons?

Minor comments:

  1. Introduction lines 73-80, authors give examples for the importance of exosomal cargo in pre-metastatic niche formation but failed to clarify the cancer types where those observations were made. Are all those studies done in osteosarcoma? Please update the examples accordingly.
  2. Discussion, lines 288-291, please include some possible explanations for the reported observations
  3. Discussion, lines 340-341, please check the sentence as it seems incomplete

Reviewer 3 Report

Mazumdar and colleagues present an excellent report, aimed at exploring the role of osteosarcoma-secreted extracellular vesicles in the establishment of a pre-metastatic niche in the lung, which would prime its microenvironment making it more prone to metastatization. The paper is well written, and the experiments are well designed. The approach taken by the authors is scientifically valid, but there are aspects which should be addressed before the authors can rule out a role for osteosarcoma-EVs in promoting the establishment of a PMN in that organ.

  • In our experience, EVs tend to accumulate in the liver, so probably the amount of EVs which got to the lungs was relatively low. I suggest to first check whether liver and spleen were more EV-positive than lungs, and in that case, to perform a pilot experiment delivering EVs directly into one of the lungs, while injecting the other with PBS: this will provide a final validation about EVs reprogramming the lung microenvironment – i.e. if the EV-injected lung develops more metastases compared to the PBS-injected lung, then the EVs are actually working to increase metastatic potential, and viceversa.
  • If the authors find that EVs accumulate significantly more in other organs compared to lungs, it would also be interesting to assess whether these sites of EV accumulation develop metastases, which would be highly unexpected and could provide valuable data if proven true
  • Another experiment that could provide valuable information is to retrieve cells directly from lung metastases, grow them back in vitro, and then isolating EVs from these lung-targeting cells and using them for an in vivo experiment, comparing them with parental 143B cells-EVs.
  • What is the effect on in vitro migration and invasion ability of 143B vs SaOS-2 EVs?This could be assessed using transwell migration and invasion assays
  • Did the authors check if EV treatment was able to affect gene expression similarly to what observed with tumor-injected mice? I suggest to check this, also in the new in vivo experiments I proposed in the earlier comments

Round 2

Reviewer 1 Report

Issues concerning the initial submission have been addressed and I would therefore recommend publication of this study.

Reviewer 2 Report

We’d like to thank Mazumda A et al for submitting their responses to the review. Based on the new data presented and edits in the manuscript, we have the following comments for the authors:

  • Figure 2 panel C: The authors have still not included the data from Control mice for Day 28 with CD11b DAPI staining. Indeed, authors have provided the quantification in Supplementary fig S3A but again the data from controls for day 28 is missing. We would appreciate it if the authors could address that.
  • Supplementary figure S3B, C: The figure legends are missing details about the time point for the representative images of CD11b and GR-1 staining. We recommend the authors to update that information
  • Lines 28-29: Recommend the authors to re-word the sentence for it to make sense grammatically
  • Lines 292-294: The authors touch upon the importance of additional pre-requisite tumor-secreted factors in PMN formation. We recommend the authors present information on these additional secretory factors and/ or include a discussion of some future studies which could help dissect PMN formation in osteosarcoma
  • Lines 343-344: This sentence still seems incomplete and is an abrupt ending to the paragraph. It will be great if the authors could summarize/ highlight the important findings from the proteomic profile of EVs isolated from a small cohort of osteosarcoma patients.
  • There are still some typographic and grammatical errors that need to be rectified.

Reviewer 3 Report

The authors did not address all the comments presented in the first round of review, therefore I would at least ask them that they dicuss the issues presented in further detail in the discussion as limitations of the study. Furthermore, in figure R3B many more cells have occupied the scratched area following EV treatment, which should be considered an increase of motility. If the authors are planning on studying aspects such as migration and invasion in more detail, I suggest them not to discuss this data and gather more evidence before stating that there is no effect.

I'm also glad that the authors found the expected regulations in gene expression in lungs following EV treatment, which will strenghten the significance of the work.

Although not perfect, in my opinion the work now only needs proofreading before being deemed fit for publication.

Round 3

Reviewer 2 Report

The authors have adequately addressed the concerns of the reviewer.